# Characterization of Alternative Splicing in High-Risk Wilms’ Tumors

**DOI:** 10.3390/ijms25084520

**Published:** 2024-04-20

**Authors:** Yaron Trink, Achia Urbach, Benjamin Dekel, Peter Hohenstein, Jacob Goldberger, Tomer Kalisky

**Affiliations:** 1Faculty of Engineering and Bar-Ilan Institute of Nanotechnology and Advanced Materials (BINA), Bar-Ilan University, Ramat Gan 5290002, Israel; trinkya@biu.ac.il (Y.T.); jacob.goldberger@biu.ac.il (J.G.); 2The Mina and Everard Goodman Faculty of Life Sciences, Bar-Ilan University, Ramat Gan 5290002, Israel; achia.urbach@biu.ac.il; 3Pediatric Stem Cell Research Institute and Division of Pediatric Nephrology, Edmond and Lily Safra Children’s Hospital, Sheba Tel-HaShomer Medical Centre, Ramat Gan 5262000, Israel; 4Department of Human Genetics, Leiden University Medical Center, 2300 RC Leiden, The Netherlands; p.hohenstein@lumc.nl

**Keywords:** Wilms’ tumors, pareto task inference, alternative mRNA splicing, cell deconvolution

## Abstract

The significant heterogeneity of Wilms’ tumors between different patients is thought to arise from genetic and epigenetic distortions that occur during various stages of fetal kidney development in a way that is poorly understood. To address this, we characterized the heterogeneity of alternative mRNA splicing in Wilms’ tumors using a publicly available RNAseq dataset of high-risk Wilms’ tumors and normal kidney samples. Through Pareto task inference and cell deconvolution, we found that the tumors and normal kidney samples are organized according to progressive stages of kidney development within a triangle-shaped region in latent space, whose vertices, or “archetypes”, resemble the cap mesenchyme, the nephrogenic stroma, and epithelial tubular structures of the fetal kidney. We identified a set of genes that are alternatively spliced between tumors located in different regions of latent space and found that many of these genes are associated with the epithelial-to-mesenchymal transition (EMT) and muscle development. Using motif enrichment analysis, we identified putative splicing regulators, some of which are associated with kidney development. Our findings provide new insights into the etiology of Wilms’ tumors and suggest that specific splicing mechanisms in early stages of development may contribute to tumor development in different patients.

## 1. Introduction

Wilms’ tumors, also referred to as nephroblastomas, are malignancies affecting the kidney that primarily occur in children below the age of five. The tumorigenesis of Wilms’ tumors is believed to be closely associated with fetal nephron development since they often contain “blastemal” cells that histologically resemble fetal nephron progenitors. Moreover, Wilms’ tumors often arise from or in the vicinity of “nephrogenic rests”—poorly differentiated structures that resemble early embryonic kidney precursor cells not usually seen in the postnatal kidney [1,2,3]. These observations suggest that genetic and epigenetic aberrations during various stages of fetal kidney development lead to Wilms’ tumor formation, and that these aberrations result in extensive heterogeneity among patients.

Wilms’ tumors are histologically and clinically diverse. Most cases are classified as favorable histology Wilms’ tumors (FHWTs), which typically contain “stromal”, “blastemal”, and “epithelial” components. The stromal component resembles the nephrogenic stroma of the developing fetal kidney and may also contain cells resembling smooth or skeletal muscle, fat, cartilage, bone, and glial cells. The blastemal component histologically resembles the cap mesenchyme, a transient cell compartment of the fetal developing kidney which contains nephron progenitors [4]. In the fetal kidney, these nephron progenitors undergo a mesenchymal-to-epithelial transition (MET) and differentiate into the various epithelial tubular segments of the nephron [5]. The epithelial component typically contains early non-functional epithelial structures from early stages of nephron differentiation. Tumors containing all three components are classified as “mixed” type or “triphasic”, whereas those containing a single dominant histology are classified as “blastemal”, “stromal”, or “epithelial” tumors. The blastemal component is presumed to be most malignant, and tumors containing a large blastemal component that survived preoperative chemotherapy are regarded as “high-risk” tumors and require more aggressive treatment [4,6,7]. Anaplastic Wilms’ tumors are characterized by multipolar mitoses, nuclear enlargement, and hyperchromatic nuclei [8], and are classified as “focal” (FAWT) or “diffuse” (DAWT) depending on the geographical distribution of the anaplastic cells within the tumor [6,9]. Diffuse anaplastic Wilms’ tumors are associated with lower event-free survival and lower overall survival estimates when compared to favorable histology Wilms’ tumors, and are also regarded as “high-risk” tumors [6,7,9]. Another class of high-risk tumors are favorable histology Wilms’ tumors that relapsed [6,10].

There are two main protocols for treating Wilms’ tumors. The Children’s Oncology Group (COG) in North America recommends initial surgery followed by chemotherapy, whereas the International Society of Pediatric Oncology (SIOP), followed by European countries, favors initial chemotherapy prior to surgery [6,11]. While efforts from both of these collaborative groups have significantly improved treatment efficacy and survival rates, considerable challenges remain. First, clinical outcomes for high-risk patients remain significantly worse when compared to low-risk patients [12]. In addition, it is crucial to improve risk stratification in order to minimize unnecessary therapeutic exposure and possible subsequent long-term complications for low-risk patients. Advances in the understanding of tumorigenesis have significant implications for this goal [13].

Previous analyses of Wilms’ tumors based on DNA microarrays or high-throughput sequencing methods have been used to identify subsets of tumors based on gene expression patterns [14,15], to find differentially expressed genes between Wilms’ tumors and normal kidney samples [16,17], and to search for genetic markers associated with relapse [18,19]. In a previous study [20], we observed that favorable histology Wilms’ tumors (treated according to the COG protocol) form a triangular shaped continuum in gene expression latent space, and that the vertices of this triangle, which represent tumor “archetypes,” have blastemal, stromal, and epithelial characteristics, corresponding to the three main lineages of the developing fetal kidney. In a consecutive study [21], we found that this geometry is also conserved in high-risk tumors that were treated with chemotherapy prior to surgery (according to the SIOP protocol) but still contained a significant amount of remaining viable blastema [15], and used a probabilistic generative model [22] to represent each tumor as a mixture of three latent biological “topics” with stromal, epithelial, and blastemal characteristics.

While these studies focused solely on gene expression in Wilms’ tumor subtypes, there is widespread evidence that alternative splicing also plays a major role in the tumorigenesis and phenotypes of tumors, many of which exhibit significant and widespread splicing abnormalities when compared to their normal counterparts [23]. The dysregulation of RNA splicing has been found to directly affect some of the hallmarks of cancer such as metastasis, one of the most difficult issues affecting cancer therapy [24]. Many alternative splicing events have been found to play a role in the epithelial-to-mesenchymal transition (EMT), whereby polarized epithelial cells undergo a series of transitions to assume a mesenchymal phenotype. This phenotype, which exhibits migratory capability and increased invasiveness, confers a metastatic ability upon cells [25,26]. For example, the skipping of exon 11 in the gene RON has been shown to enhance tumor migratory abilities [27], and splice isoform switching from the epithelial to mesenchymal isoform of the gene FGFR2 has been shown to induce EMT [28,29,30,31,32]. Likewise, RNA binding proteins known to regulate mRNA splicing were found to be abnormally expressed or somatically mutated in cancer [33,34,35,36,37]. The RNA binding protein epithelial splicing regulatory protein 1 (ESRP1), for example, has been shown to regulate the splicing of CD44, and knockdown of ESRP1 was shown to suppress lung cancer metastasis [38]. Specifically in Wilms’ tumor, a recent study showed that the splicing regulator ESRP2 is repressed by DNA methylation, whereas the overexpression of ESRP2 in Wilms’ tumor cell lines promotes alternative splicing and inhibits cell proliferation both in vitro and in vivo [39].

In a recent work, we also found that fetal kidney cells in an early developmental stage have a mesenchymal splice-isoform profile that is similar to that of blastemal-predominant Wilms’ tumor xenografts [40]. However, a comprehensive analysis of alternative splicing in high-risk Wilms’ tumors is lacking, and could provide insights into the molecular processes driving tumor development and their relation to kidney development. It could also provide insights into the functional role of alternative splicing events, specifically with regard to the possible role of EMT in establishing different tumor phenotypes. Finally, such an analysis could pave the way for targeting biomarkers for tumor subtypes, an urgently needed clinical effort [41,42].

In this study, we set out to comprehensively characterize alternative splicing in high-risk Wilms’ tumors. We first downloaded a publicly available dataset consisting of RNA sequences collected from favorable histology Wilms’ tumors (FHWTs) that relapsed, diffuse anaplastic Wilms’ tumors (DAWT), and associated normal kidney samples [10]. These tumors were treated according to the COG protocol and collected by the NCI TARGET (“Therapeutically Applicable Research to Generate Effective Treatments”) initiative. Using Pareto task inference [43], Gene Ontology enrichment analysis, and cell deconvolution [44], we found that the tumors and normal kidney samples are organized according to progressive stages of kidney development within a triangle-shaped region in latent space, whose vertices, or “archetypes”, correspond to cell states resembling the cap mesenchyme, the nephrogenic stroma, and epithelial tubular structures of the fetal kidney. We next identified genes that are alternatively spliced between samples located near the three archetypes and found that many of these genes are related to EMT and muscle structure and development. Finally, we used motif enrichment analysis for known RNA binding proteins to identify putative splicing regulators, some of which were previously found in kidney development. We anticipate that these findings will contribute to a better understanding of the role of alternative mRNA splicing in the formation of Wilms’ tumors in diverse patient populations.

## 2. Results

### 2.1. High-Risk Wilms’ Tumors and Normal Kidney Samples Form a Triangle-Shaped Continuum in Latent Space That Is Bounded by Archetypes with Blastemal, Stromal, and Epithelial Characteristics

We first downloaded an RNA sequencing dataset from the NCI TARGET [10] study containing 130 Wilms’ tumor samples and six associated normal kidney samples. The tumors were all classified as “high-risk” since they were either relapsed favorable histology Wilms’ tumors (FHWTs) or diffuse anaplastic Wilms’ tumors (DAWTs). We then performed sequence alignment and obtained a gene expression matrix (Appendix A). Using Principal Component Analysis (PCA), we found that high-risk Wilms’ tumors form a continuum in gene expression latent space rather than discrete well-separated clusters [20,21] (Figure 1A–F, Appendix A). To better understand this continuous heterogeneity, we used Pareto task inference [43,45] to calculate the vertices of the best fitting polytope encompassing all tumors and normal kidney samples in latent space. These vertices, or “archetypes”, presumably represent idealized cell types from which all samples within the continuum are composed, where the precise transcriptional state of each sample determines its position in latent space relative to each biological archetype [20,43,46]. This technique allows us to summarize the gene expression profile of each sample as a combination of different archetypal, or pure, biological states, based on a theoretical evolutionary principle [47]. We tested different numbers of archetypes and found that the biological heterogeneity of our dataset could be described by both three and four archetypes. However, closer inspection of the fourth archetype showed that most of the variability captured by this archetype is likely related to RNAseq library size (Appendix A). We therefore decided to focus on the polytope summarized by three archetypes which best summarize the biological heterogeneity of our dataset.

To infer the biological identity of the three archetypes, we checked the expression levels of selected genes that are known to mark the different lineages in the developing kidney (Figure 1A–C and Figure 2A). We found that the gene SALL2, a marker for the cap mesenchyme, is overexpressed near the first archetype; hence, we named it the “blastemal” archetype. Similarly, the gene COL3A1, which is a marker of the nephrogenic stroma, is overexpressed in the vicinity of the second archetype, which we named the “stromal” archetype, and the gene UMOD, which is a marker for renal epithelial cells, is overexpressed in the normal samples near the third archetype, which we named the “normal” archetype. We also observed that MYOG, a muscle-specific transcription factor with myogenesis-inducing capabilities, and MYL1, a motor protein expressed in muscle cells, are overexpressed near the stromal archetype (Figure 1D,E). Consistent with the findings of the original paper by the TARGET initiative [10], we did not observe an overall significant separation in gene expression latent space between tumors with favorable histology (FHWT) and diffuse anaplasia (DAWT) (Figure 1F).

We further characterized the identity of the archetypes by performing Gene Ontology (GO) enrichment analysis (Figure 1G, Appendix A). We chose genes that are overexpressed (log2FC > 3) in each of the three archetypes with respect to the other two and used these as inputs to ToppGene [48]. We observed that genes that are overexpressed in the blastemal archetype are related to the cap mesenchyme and nephron progenitors, genes that are overexpressed in the stromal archetype are related to the nephrogenic stroma (the un-induced mesenchyme) and muscle development, and genes that are enriched in the normal archetype are related to more differentiated epithelial structures of the nephron such as the proximal tubule, the loop of Henle (LOH), and the collecting duct.

### 2.2. High-Risk Wilms’ Tumors Are Organized in Latent Space According to Progressive Stages of Kidney Development

In order to better understand the relationship between Wilms’ tumor heterogeneity and kidney development, we performed hierarchical clustering using expression levels of 102 selected genes that are known from the literature to be involved in kidney development (Figure 2A, Appendix A). We observed that, indeed, the tumors near the stromal archetype have a high expression of genes related to the nephrogenic stroma (e.g., COL3A1, COL5A2, and VIM) and muscle structure and development (e.g., DES, MYL1, and MYOG), and that tumors located near the blastemal archetype have a high expression of genes marking the cap mesenchyme (e.g., CITED1, EYA1, and SALL2). Likewise, we observed that the normal samples have a high expression of renal epithelial markers, mainly for the loop of Henle and the distal tubule (e.g., CDH1, KCNJ1, MUC1, SLC12A1, and UMOD). Finally, we identified six “epithelial-like” tumors that are located distinctly from the others in latent space (marked “Epithelial 1” and “Epithelial 2” in Figure 2B) and that also express some renal epithelial markers (e.g., CDH6, KRT18, and KRT19, Appendix A).

The bulk expression profiles in this dataset are the average expression of the cell types present in the sample. To enhance the resolution of our analysis, we next used cell deconvolution [44] to predict the proportions of different cell types in each sample. Since Wilms’ tumors contain cells resembling those of the fetal kidney, we used a publicly available atlas of single-cell gene expression from the human fetal kidney [49] as a reference. After deconvolving each sample, we plotted the proportions of each predicted fetal cell subpopulation (e.g., “fibroblast” and “proximal tubule”) for each sample in gene expression latent space (Figure 2B and Appendix A). We observed that the tumors are arranged according to progressive developmental stages in latent space. For example, high proportions of cells resembling the cap mesenchyme are found within tumors located near the blastemal archetype, whereas cells resembling early stages of nephron differentiation (“renal vesicle” and “S-shaped body”) are found in progressively higher proportions in the more epithelial-like tumors that are located towards the normal archetype (marked as “Epithelial 1” and “Epithelial 2” in Figure 2B). Cells resembling fully differentiated epithelial structures (“proximal tubule” and “loop of Henle”) were found to be predominant in the normal samples. Likewise, we observed that tumors located near the stromal archetype have a high proportion of cells resembling renal fibroblasts. We performed similar analysis on a dataset of high-risk blastemal-type Wilms’ tumors (treated according to SIOP protocols) published by Wegert et al. [15] and found similar results [21] (see a detailed comparison in Appendix A).

### 2.3. A Significant Number of RNA Transcripts Related to EMT and Muscle Development Are Alternatively Spliced between High-Risk Wilms’ Tumors Located in Different Regions of Latent Space

We next set to identify genes that are alternatively spliced between the different tumors and normal kidney samples located in different regions of latent space. We first chose three groups of samples, each containing five samples located near one of the three archetypes (Appendix A). We then used rMATS [50] to perform three comparisons between these three groups. The results returned by rMATS (Figure 3A,B; Appendix A) include a set of transcripts that are significantly (FDR = 0) alternatively spliced between tumors representing each of the three transcriptional archetypes. Gene Ontology (GO) enrichment analysis for alternatively skipped exons (FDR = 0, |∆ψ|>0.1) showed that a significant fraction of these transcripts are related to the epithelial-to-mesenchymal transition (EMT) and muscle development (Figure 4A, Appendix A).

For example, the gene FGFR2, a gene that is known to have epithelial and mesenchymal splice variants [28,51,52,53], is expressed in its mesenchymal form in the majority of tumors. Similarly, the gene PRMT2, a gene that is involved in cancer invasion, growth, and presumably EMT, and whose splice variants have been linked to cancer [54,55], contains an exon that is overexpressed near the stromal and normal archetypes, as well as some of the epithelial-like tumors. Likewise, the gene TPM2, whose splice variants and differential expression have been associated with EMT [56,57], cancer [58,59], and muscle diseases [60], is expressed in its muscle-specific isoform [58,61] near the stromal archetype. Another example is the gene FLNB whose exon 30 was found to be skipped in the majority of tumors (Appendix A), which is consistent with previous findings that the skipping of this exon induces EMT, is associated with basal-like breast cancer, and is regulated by the RNA binding proteins QKI and RBFOX1 [62]. Similarly, in the gene ATP2B1 (PMCA1), an isoform that is overexpressed specifically in skeletal and heart muscle but not in kidney tissues [63,64], was found to be overexpressed near the stromal archetype (Appendix A). Another notable example is the gene MEF2D [65], whose muscle-specific isoform is overexpressed near the stromal archetype (Appendix A). The muscle-specific isoform for this gene was previously found to be promoted by the splicing regulators RBFOX1/2 [66,67].

### 2.4. Motif Enrichment Analysis Reveals Putative RNA Binding Proteins (RBPs) Regulating Alternative Splicing between Wilms’ Tumors Located in Different Regions of Latent Space

Alternative RNA splicing is controlled by RNA binding proteins (RBPs) which bind to pre-mRNA upstream or downstream of the regulated exons. GO enrichment analysis for transcripts that are alternatively spliced between samples located near the three archetypes suggests that many of them are known targets for the splicing regulators RBFOX2, ESRP1, and ESRP2 (Figure 4A). Therefore, in order to identify more putative splicing regulators, we used rMAPS [68,69] to perform enrichment analysis for RNA binding motifs belonging to known RNA binding proteins (Figure 4B,C; Appendix A). We found a set of putative splicing regulators whose binding motifs are enriched upstream or downstream of exons that are alternatively spliced, and that are also differentially expressed between the samples in different locations of latent space. The most prominent motif enrichment was found for the RNA binding proteins RBFOX1 [62,70] and QKI [62,70,71], which indicates that in the more stromal tumors, these splicing regulators are overexpressed and tend to bind downstream of their target exons and promote their inclusion. We note that RBFOX1 and QKI were previously shown to physically interact with each other in order to regulate alternative splicing in mammary epithelial cells, thereby promoting an epithelial-to-mesenchymal transition (EMT) [62], and have also been associated with the mesenchymal-to-epithelial transition (MET) that occurs during fetal kidney development [40].

## 3. Discussion

In this study, we characterized the continuous heterogeneity of high-risk Wilms’ tumors using gene expression and mRNA splicing. We first showed that high-risk Wilms’ tumors and normal kidney samples from different patients form a continuum in gene expression latent space that is bound by stromal, blastemal, and epithelial archetypes that resemble the nephrogenic stroma (the un-induced metanephric mesenchyme), the cap mesenchyme, and the early epithelial structures of the fetal kidney, respectively. Our previous studies have yielded archetypes with similar identities for blastemal-type Wilms’ tumors treated according to SIOP protocols [21], and for a meta-analysis of hundreds of Wilms’ tumors from a series of studies by the COG [20]. Our results here show that the identity of the archetypes is conserved also for an additional set of high-risk Wilms’ tumors [10] (DAWT and relapsed FHWT). We next explored the alternative mRNA splicing landscape of high-risk Wilms’ tumors and found that many transcripts in these tumors are alternatively spliced in different regions of gene expression latent space. Moreover, we found that a significant fraction of these transcripts are associated with EMT and muscle development. In particular, we observed an elevated expression of muscle-specific isoforms in tumors located near the stromal archetype, which we did not discuss in previous studies [20,21].

The development of Wilms’ tumors has been tightly linked to aberrations in fetal kidney development. Two central processes in embryonic nephrogenesis are the induction of the metanephric mesenchyme, resulting in the condensation of cells from the nephrogenic stroma around the ureteric tip to form the cap mesenchyme, and the mesenchymal-to-epithelial transition (MET), by which cells from the cap mesenchyme progressively differentiate through a series of transformations to form the various epithelial segments of the nephron. Both the kidney metanephric mesenchyme and muscle tissues originate from the fetal mesoderm, but whereas the kidney is generally accepted to be derived from the intermediate mesoderm, muscles are paraxial-mesoderm-derived. Interestingly, although much is known about the intermediate mesoderm origin of the nephrogenic lineage, the origin of the stromal lineage is much less clear, and some lines of evidence even suggest that the renal stroma could be derived from the paraxial mesoderm [72].

The finding of elevated muscle-specific isoforms in tumors near the stromal archetype is therefore also relevant for the identification of the developmental stage and cell type of origin of Wilms’ tumors. Generally, Wilms’ tumors are believed to originate from the nephron progenitor cells in the cap mesenchyme, explaining the blastemal and epithelial components found in Wilms’ tumors [2]. Stromal components in the tumors might be more difficult to explain from these cells as the cell of origin. Possibly, different types of Wilms’ tumors have different stages and cell-types of origin, leading to their different histological characteristics. A subset of tumors shows clearly recognizable ectopic muscle development, and these are usually *WT1*-mutant cases [73,74]. Data from mouse models [75], human iPSC-derived organoids [76], and a recent analysis of bilateral Wilms’ tumors (also often *WT1*-mutant; [77]) all support an earlier stage of origin of the tumors than other cases, for instance, before or at the stage where the intermediate and paraxial mesoderm separate. If the renal stroma would indeed be of paraxial origin, this would offer a different developmental trajectory for these tumors. It should be noted, however, that *WT1*-mutant cases are favorable histology Wilms’ tumors, not high-risk cases as described in the present study. A final possibility to explain the finding of muscle-specific isoform expression in tumors near the stromal archetype could be that after initial differentiation into an early kidney lineage, they had undergone dedifferentiation or differentiation to a muscle-like lineage, or even direct transdifferentiation from the nephrogenic lineage to a stromal identity with more characteristics of early myoblasts. A comparable shift has been observed in the nephron progenitor cell-specific loss of *Pax2* [78]. Future functional studies will be required to test these possibilities in the context of Wilms’ tumor mutations.

Our analysis allowed us to model each tumor as a convex mixture of archetypal gene expression profiles, each of which reflects an “extreme” or “idealized” biological state. The observed configuration of the tumors in gene expression latent space allowed us to infer splicing differences between neighbors of these archetypal tumors. This analysis makes a strong assumption, namely that each tumor can be effectively summarized as a simple combination of k different archetypes, where k is a subjective hyperparameter. The number of chosen archetypes must reflect the heterogeneity and complexity of the data while remaining sufficiently low as to allow the interpretation of each archetype’s biological identity with confidence. We found that using either three or four archetypes allowed us to interpret each archetype confidently.

We note that in our study, we analyzed a small sample size of 130 tumors, and that the tumors within this study were restricted to high-risk cases as defined by the COG treatment protocol. Our dataset also included only five recurrent samples, limiting the power of a differential analysis between primary and recurrent patients.

We also note that we used a fetal kidney single-cell dataset for the deconvolution procedure. A recent analysis generated single-cell datasets from Wilms’ tumor patients to examine the differences in cellular signals between treatment-naïve and post-chemotherapy Wilms’ tumors [79]. However, such single-cell Wilms’ tumor datasets are few and do not fully reflect the transcriptional heterogeneity between Wilms’ tumor patients.

Recent studies have investigated the possible clinical implications for the link between alternative splicing and cancer [80]. First, alternatively spliced variants have the potential to be used as biomarkers for cancer diagnosis [81]. Second, there are potential applications for the targeted treatment of aberrantly spliced RNA to restore the normal function of impacted genes. For example, a recent study examined neurons affected by the group of neurodegenerative diseases known as TDP-43 proteinopathies (such as ALS) [42]. Depletion of the RNA binding protein TDP-43 from the nucleus can lead to the expression of aberrant RNA transcripts of the gene STMN2 which exhibit so-called cryptic exons, a class of exons found within noncoding intronic regions whose inclusion into mature RNA can lead to a premature stop codon associated with these diseases. The authors reported that treatment with antisense oligonucleotides (ASOs) restored normal RNA splicing, STMN2 expression, and axonal regeneration capacity in cultured motor neurons, suggesting the potential for therapeutic intervention aimed at targeting cryptic exons to mitigate RNA mis-splicing and its associated diseases [82]. Other studies have investigated potential applications for targeted treatment using a variety of new technologies to inhibit the splicing of aberrantly spliced genes [83], to affect the function of splicing factors [84], and to design novel antigens not recognized by the immune system [85,86]. We believe that our results may help facilitate the development of novel therapies aimed at targeting specific mRNA splicing mechanisms associated with Wilms’ tumors, as well as assist in pinpointing the developmental trajectory of Wilms’ tumors in different patients.

## 4. Methods

### 4.1. Datasets and Preprocessing

A total of 136 BAM files from the TARGET Wilms’ Tumor study [10] were downloaded from the Genomics Data Commons (GDC) data portal using the GDC Data Transfer Tool Client (accessed on 1 December 2020). Each BAM file was sorted and converted to a paired-end fastq file using SAMtools (version 1.19.2) [87] and realigned to a reference genome (hg38) using STAR (version 2.7.3a) [88] to produce a gene expression counts matrix. Clinical data were manually downloaded from the GDC website. The raw gene expression counts were normalized using DESeq2 (version 1.40.2) [89] and then a modified log-transform was performed (log2(1 + x)). Genes with zero counts in all samples were removed from the analysis.

### 4.2. Data Visualization, GO Enrichment Analysis, and Clustering

PCA was performed using the “Scikit-learn” python package (version 1.2.2) [90] and PCA plots were drawn using the Matplotlib python package (version 3.7.1) [91]. Gene Ontology enrichment analysis was performed using ToppGene [48]. Hierarchical clustering was performed using the “ComplexHeatmap” R package (version 2.16.0) [92], with standardized rows (=genes). For the clustering of gene expression, we used the Pearson correlation distance metric with complete linkage. For the clustering of mRNA splicing inclusion levels, we used the Euclidean distance metric with average linkage.

### 4.3. Archetype Analysis

The “ParTI_lite” MATLAB function (https://www.weizmann.ac.il/mcb/UriAlon/download/ParTI, last accessed on 22 May 2023) [43] was used to find the best fitting polytope encompassing the data points in latent space, where each data point is the normalized gene expression vector of a specific sample. The vertices of the best fitting polytope represent archetypes—idealized tumors/tissues or cell types which specialize in a particular biological task. The “performance” of each sample in this task is determined by its geometrical distance from each of the archetypes, and the identity of the biological tasks that characterize each archetype can be inferred from genes enriched in a nearby latent space [43].

### 4.4. Cell Deconvolution

For cell deconvolution, we used the CPM [44] algorithm as implemented in the “scBio” R package (version 0.1.6). We used the human fetal kidney single-cell RNAseq dataset from the Kidney Cell Atlas [49] as a reference single-cell gene expression matrix and a UMAP [93] embedding of this dataset as the cell state space. We used the default values for the “modelSize”, “minSelection”, and “neighborhoodSize” parameters, and we set the parameter “quantifyTypes” = T, which instructs the algorithm to quantify the proportions of different reference cell sub-populations in each bulk sample, in addition to the abundance values of each reference single-cell.

### 4.5. Alternative Splicing and RNA Binding Motif Enrichment Analysis

We used rMATS (version 4.0.2) [50] to find mRNA splicing events with significant inclusion-level differences between groups of samples located near each of the archetypes. We manually inspected top alternatively spliced transcripts in the IGV genome browser [94]. We then used the tables output by rMATS as input to rMAPS (version 2.2.0) [68,69] in order to identify putative splicing regulators by searching for binding motifs belonging to known RNA binding proteins (RBPs) that are also enriched in the vicinity of alternatively spliced exons. Additionally, in order to draw the heatmap of inclusion levels for all samples in the dataset, we reran rMATS for all samples with the command-line option “–cstat 0”.

The list of RNA binding proteins was obtained from the rMAPS website (http://rmaps.cecsresearch.org/Help/RNABindingProtein, accessed on 14 April 2024). In addition to the RNA binding motifs that were tested using the default settings on the rMAPS website, we also conducted tests on additional UGG-enriched motifs that have been previously identified as binding sites for the RNA binding proteins ESRP1 [95,96] and ESRP2 [97] (Appendix A). Following Yang et al. [71] and the CISBP-RNA database [98] (http://cisbp-rna.ccbr.utoronto.ca, accessed on 14 April 2024), we assumed that the proteins RBFOX1 and RBFOX2 both bind to the same mRNA motif ([AT]GCATG[AC]).

## Figures and Tables

**Figure 1 ijms-25-04520-f001:**
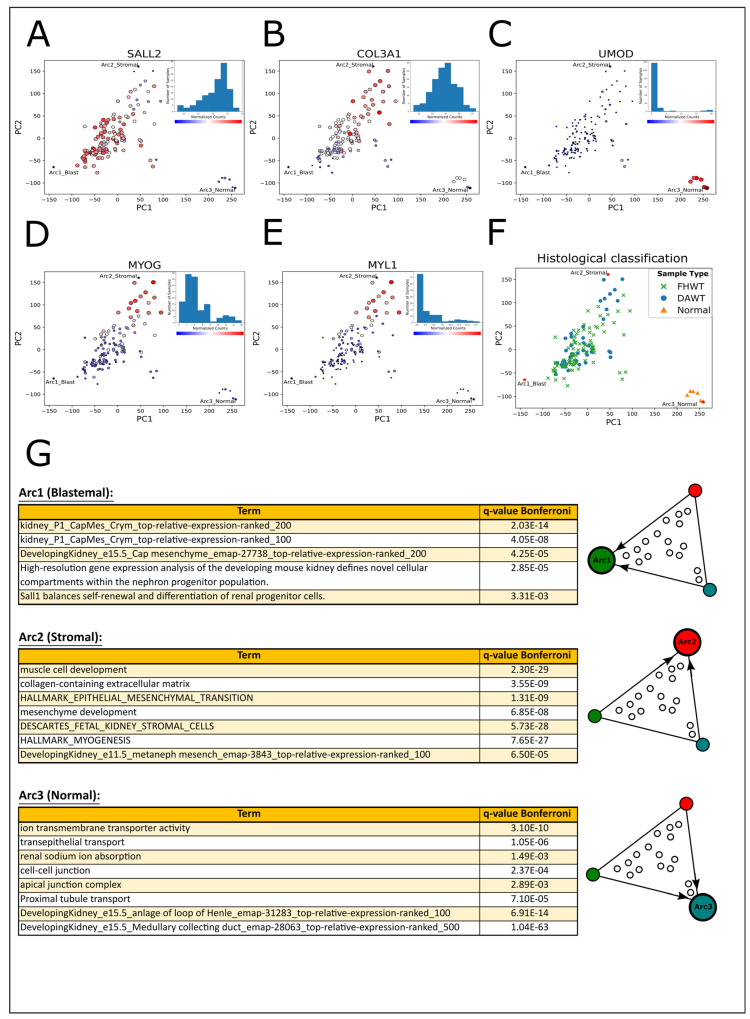
High-risk Wilms’ tumors and normal kidney samples form a triangle-shaped continuum in latent space that is bounded by archetypes with stromal, blastemal, and epithelial characteristics that resemble the nephrogenic stroma, the cap mesenchyme, and early epithelial structures of the fetal kidney, respectively. (**A**–**C**) Shown are PCA plots of RNAseq gene expression profiles from 130 high-risk Wilms’ tumors (relapsed favorable histology or diffuse anaplastic Wilms’ tumors) and six normal kidneys from the US NCI TARGET project. Three archetypes, denoted in the plots by asterisks and labeled “Arc1_Blast”, “Arc2_Stromal”, and “Arc3_Normal”, form the vertices of the best fitting polytope which encompasses all data points. To identify the archetypes, the size and color of each sample (dot) are drawn according to expression levels of known genes marking the cell populations of the fetal developing kidney (large red—high expression; small blue—low expression). Histograms of gene expression levels are included in each PCA subplot. The gene SALL2, which marks the cap mesenchyme, is highly expressed near Arc1, which is the “blastemal” archetype. The gene COL3A1, which marks the nephrogenic stroma, is highly expressed near Arc2, which is the “stromal” archetype. Likewise, UMOD, a gene which is highly expressed in the epithelial cells of the loop of Henle, is highly expressed in the normal samples located near Arc3, which is the “normal” archetype that is predominantly epithelial. (**D**,**E**) MYOG, a transcription factor that can induce myogenesis, and MYL1, a gene that encodes a muscle motor protein, are highly expressed near the “stromal” archetype (Arc2_Stromal). (**F**) A PCA plot with each sample marked according to its histological classification. We did not observe a significant separation between relapsed favorable histology Wilms’ tumors (FHWTs) and diffuse anaplastic Wilms’ tumors (DAWTs) in this dataset. (**G**) Genes overexpressed (log2FC > 3) in each of the three archetypes with respect to both of the other two were used as input for Gene Ontology enrichment analysis. Arc1, the “blastemal” archetype, overexpresses genes related to the cap mesenchyme and nephron progenitor cells. Arc2, the “stromal” archetype, overexpresses genes that are related to the extracellular matrix, the nephrogenic stroma (the un-induced metanephric mesenchyme), and muscle development. Arc3, the “normal” archetype, overexpresses genes that are related to the structure and function of epithelial lineages of the kidney.

**Figure 2 ijms-25-04520-f002:**
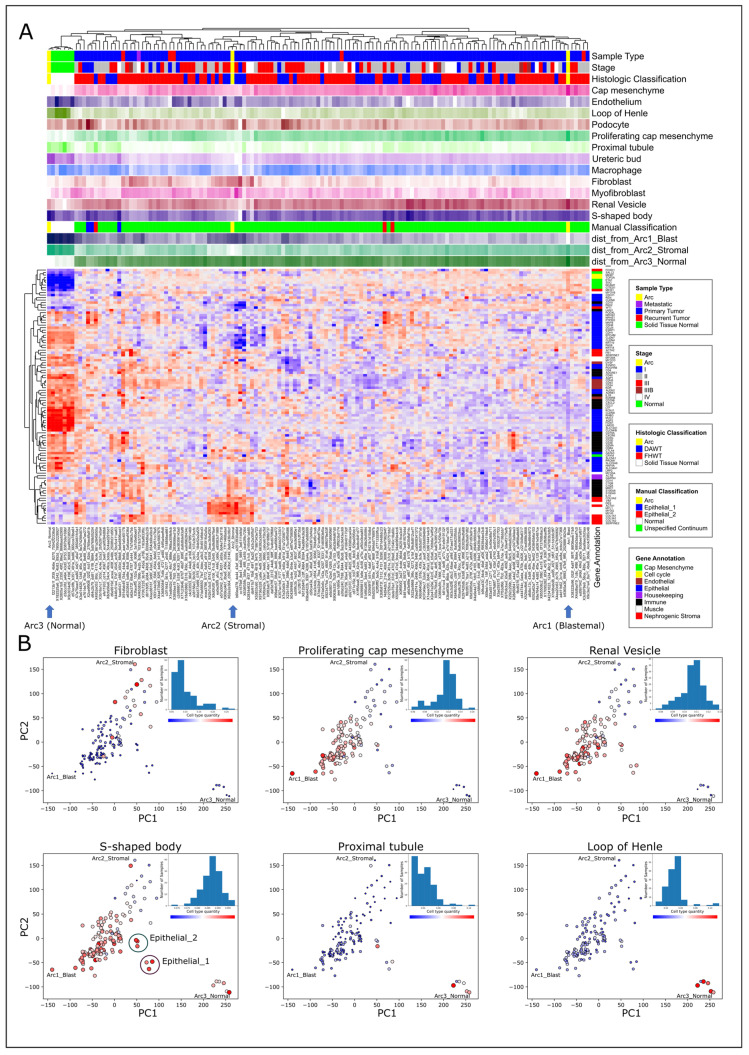
Cell deconvolution shows that high-risk Wilms’ tumors are organized in latent space according to progressive stages of kidney development. (**A**) A gene expression heatmap of 102 selected genes that are known from the literature to be involved in kidney development. The top panels show the clinical and histological information for each sample, the proportions of cell types within each sample as predicted by cell deconvolution, and the distances from each of the three archetypes (for cell deconvolution and distances to archetypes: light color—low; dark color—high). It can be seen that genes characteristic to the cap mesenchyme (e.g., CITED1, EYA1, and SALL2) are overexpressed near the blastemal archetype. Genes characteristic of the nephrogenic stroma (e.g., COL3A1, COL5A2, and VIM), as well as muscle structure and development (e.g., DES, MYL1, and MYOG), are overexpressed in the vicinity of the stromal archetype. Likewise, genes marking epithelial tubular structures, mainly the loop of Henle and the distal tubule (e.g., CDH1, KCNJ1, MUC1, SLC12A1, and UMOD), are overexpressed near the normal archetype. (**B**) PCA plots marking proportions of fetal kidney cell populations within each sample, as inferred by cell deconvolution. The size and color of each data point are drawn according to the proportion of the selected population from the predicted cellular composition (large red—high proportion; small blue—low proportion). A histogram of the predicted cell type proportions for all cells is included in each PCA plot. It can be seen that the samples are arranged in latent space according to progressive stages of the developing kidney: cells resembling the cap mesenchyme are predominant in tumors located near the blastemal archetype, whereas cells resembling early nephron differentiation such as the renal vesicle and S-shaped body are found at progressively higher proportions in the more epithelial-like tumors that are located towards the normal archetype (“Epithelial 1” and “Epithelial 2”). Similarly, cells resembling fully differentiated epithelial structures such as the proximal tubule and the loop of Henle are predominant in the normal samples. Likewise, we observed that tumors located near the stromal archetype have a high proportion of cells resembling renal fibroblasts, which are the predominant component of the nephrogenic stroma.

**Figure 3 ijms-25-04520-f003:**
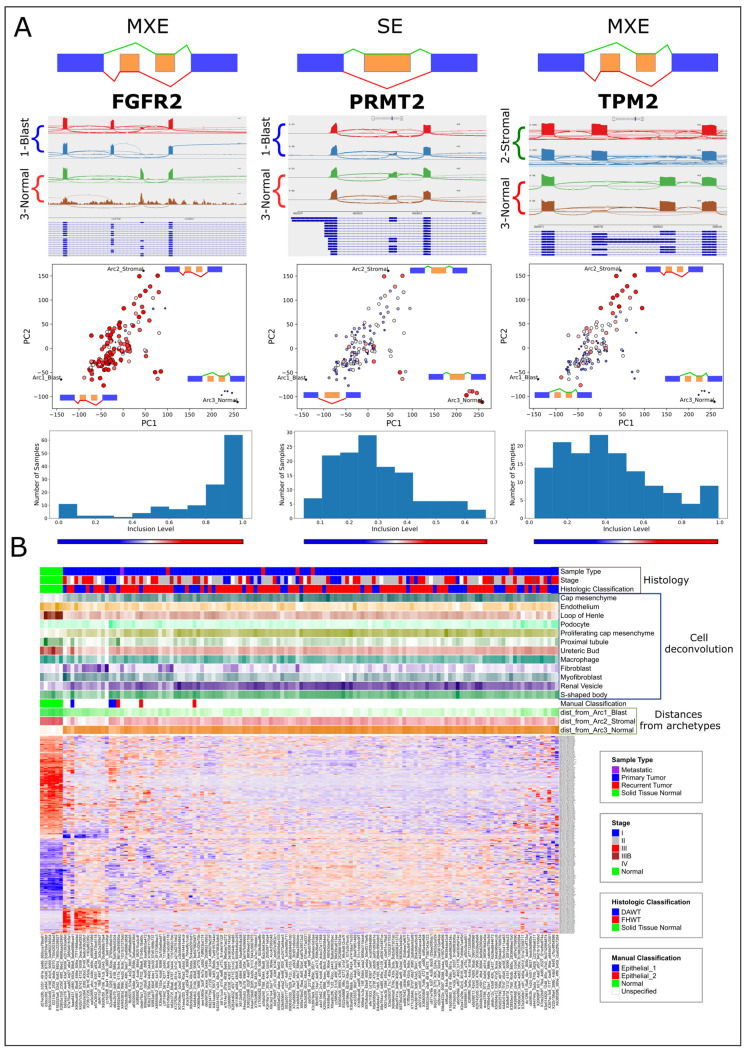
RNA transcripts related to EMT and muscle development are alternatively spliced between high-risk Wilms’ tumors located in different regions of latent space. (**A**) Sashimi plots (top), feature plots (middle), and inclusion-level histograms (bottom) for the alternatively spliced genes FGFR2, PRMT2, and TPM2. The sashimi plots show representative samples that are located near the archetypes (denoted by asterisks). The feature plots show the isoform inclusion levels in the different tumors and samples in latent space, where the size and color of each sample are drawn according to the inclusion levels of a particular isoform (large red—high; small blue—low). (**B**) A heatmap showing the inclusion levels of 277 significantly skipped/included exons (SEs) and mutually exclusive exons (MXEs) for transcripts that were found to be significantly alternatively spliced in latent space. The rows are the union of all significant alternative splicing events (FDR = 0, |∆ψ|>0.2) from each of the three comparisons between the five nearest neighbors of the three archetypes. We plotted only SE (single-exon) and MXE (mutually exclusive exon) splicing events since these were the majority. The top panels show clinical and histological information, the proportions of cell types in each sample as predicted by cell deconvolution, our manual classification based on tumor location in latent space, and the distances to each of the three archetypes (for cell deconvolution and distances to archetypes: light color—low; dark color—high).

**Figure 4 ijms-25-04520-f004:**
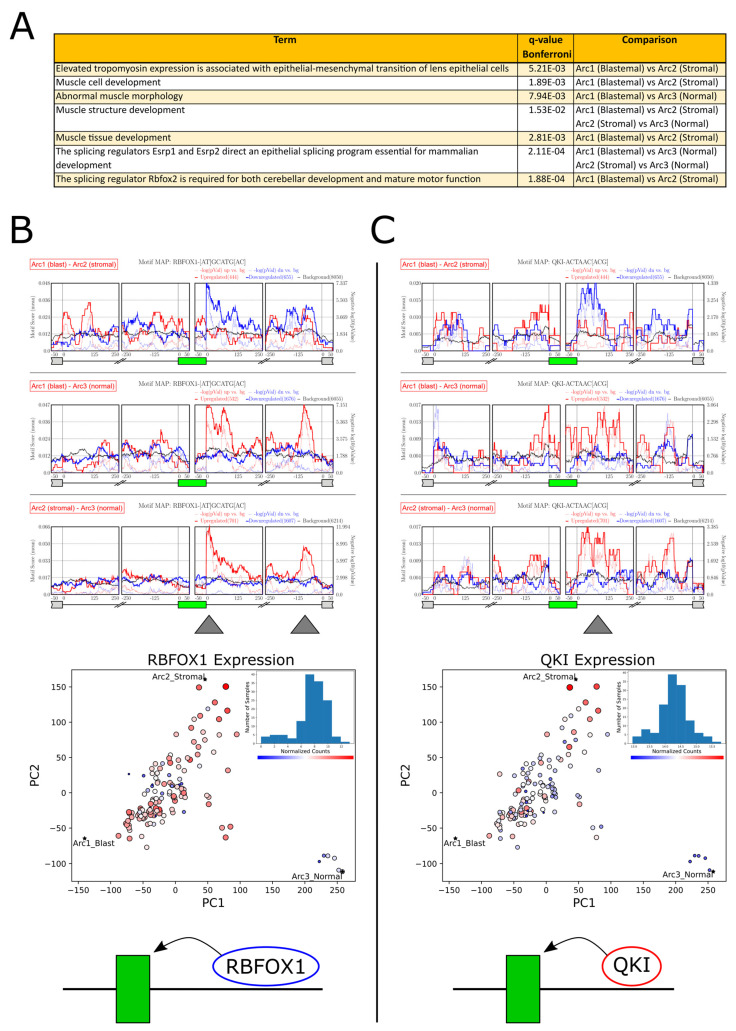
Motif enrichment analysis reveals putative RNA binding proteins (RBPs) regulating alternative mRNA splicing between Wilms’ tumors located in different regions of latent space. (**A**) Gene Ontology (GO) enrichment analysis for exons that were found to be significantly alternatively spliced (FDR = 0, |∆ψ|>0.1) in latent space, as identified from the three comparisons between the five nearest neighbors of the three archetypes. A significant fraction of these transcripts are related to the epithelial-to-mesenchymal transition (EMT) and muscle development. (**B**,**C**) Motif enrichment diagrams (top) and expression feature plots (middle) for the splicing regulators RBFOX1 and QKI. Histograms of RBFOX1 and QKI expression levels are included in each PCA subplot (archetypes denoted by asterisks). It can be seen that the expression levels of these RNA binding proteins are elevated in the blastemal, and even more in the stromal tumors. Independently, we observed that exons that are elevated in these tumors are enriched for binding motifs of these regulators at their downstream introns. This indicates that in these tumors, RBFOX1 and QKI bind to the mRNA downstream of the mesenchymal-associated cassette exons and promote their inclusion (bottom sketch).

## Data Availability

The results published here are in whole or in part based upon data generated by the Therapeutically Applicable Research to Generate Effective Treatments (TARGET) initiative, phs000218, managed by the NCI. The data used for this analysis are available through the GDC (https://gdc.cancer.gov/about-data/publications#/?groups=&years=&programs=TARGET&order=desc). Information about TARGET can be found at https://www.cancer.gov/ccg/research/genome-sequencing/target/about, both accessed on 15 November 2020.

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
