# Peer review of "Characterization of Alternative Splicing in High-Risk Wilms’ Tumors"

_ijms, 2024, doi:10.3390/ijms25084520_

Round 1
Reviewer 1 Report
Comments and Suggestions for Authors
The study is interesting and analysis is well performed, however, the same kind of analysis (in particular the one referring to gene expression) has been done by the same authors on another kind (blastemal) of high-risk WT and on favorable histology WT samples. Therefore, its novelty is limited to: 1) the analysis of the splicing motifs distribution in the selected tumor samples, and the association of the identified transcript with EMT and muscle development; 2) the identification of putative splicing regulators related to identified splicing motifs. Regarding this second point, only two of the splicing regulators, RBFOX1 and QKI, are cited. Functional studies evaluating their activity on selected target genes in different WT samples would have improved the significance of the paper.
Regarding the Discussion, this is a mere recapitulation of data with considerations that have already been made throughout the manuscript. Thus, it should be revised by broadening the discussion including also results described in previous papers (for example, see cited ref. 12, 19, and 20), in order to give a more detailed overview of the available evidence about the issue. In addition, the last part concerning future studies (aimed at evaluating hypotheses about stromal WT origin) and the impact that the characterization of alternative mRNA splicing patterns/mechanisms may have on the design of novel therapeutic approaches against WT, should be more widely discussed.
Reviewer 2 Report
Comments and Suggestions for Authors
Overall, the manuscript is highlighting the importance of studying alternative splicing in Wilms' tumors. However, it could be strengthened by
1. In the introduction, explicitly describing the splicing events to specific challenges in diagnosing, treating, or predicting the prognosis of high-risk Wilms' tumors.
2. Stating clear hypothesis or predictions based on the background information in the introduction. This will sharpen the research focus and set clearer expectations for the reader.
3. Highlighting specific clinical or therapeutic dilemmas would underscore the urgency and relevance of the research.
4. Tightening the focus of the introduction on alternative splicing's role in cancer, specifically Wilms' tumors. Consider including a more detailed discussion of known splicing mechanisms and their implications for tumor biology.
Here are some additional suggestions for specific sections:
Figures:
· In Figure 2B, consider merging identical x and y-axis labels and increasing the combined label size for better clarity.
· Additional visual aids or detailed explanations, especially for Figures 3 and 4, could help readers understand the spatial relationships and splicing events described.
· Results: Briefly commenting on the biological implications of the findings within this section could provide immediate context and set the stage for a more detailed discussion later.
· Discussion: Elaborate on the choice of three archetypes in the PCA and Pareto Task Inference analysis and the rationale for not including a fourth due to RNAseq library size variability.
· Briefly discuss the potential biological significance of the observed patterns and their statistical robustness.
· Mention specific limitations of the study, such as dataset limitations, potential biases, or limitations of the computational models used.
· Mention the specific versions of software tools and packages used for reproducibility purposes.
· Briefly explain the rationale for key methodological choices, such as the selection of clustering algorithms or reference datasets for cell deconvolution.
Comments on the Quality of English LanguageThe text necessitates a moderate level of revision and refinement to enhance its clarity, coherence, and overall proficiency in the English language.
Reviewer 3 Report
Comments and Suggestions for Authors
1. The sample type of Figure 1-E should be included, and the expression scale for each gene is required to be added.
2. Supplemental information are suggested to be integrated into single file, such as figures and tables. Further, the citation of supplemental data in the main text should be precisely annotated.
line 132-134 “whereas including an additional fourth archetype in the 3rd dimension 132 captures variability that is likely related to RNAseq library size (see supplementary information), it is suggested to modify “see supplementary information” to “Supplemental Fig. s …”
3. Despite the explanation in the legends, line 154-166, it is still lack of highlighting the significance of the classification into three archetypes in the matin text. It is necessary to elucidate the basics, hypotheses, and annotations for the subtype classification.
“Genes overexpressed (log2FC > 3) in each of the three archetypes with respect to the other two were used as input for gene ontology enrichment analysis. Arc1, the “blastemal” archetype, overexpresses genes related to the cap mesenchyme and nephron progenitor cells. Arc2, the “stromal” archetype, overexpresses genes that are related to the extracellular matrix, the nephrogenic stroma (the un-induced metanephric mesenchyme), and muscle development. Arc3, the “normal” archetype, overexpresses genes that are related to the structure and function of epithelial lineages of the kidney.”
Further, the “Figure S11: A fourth archetype highlights tumor samples with smaller RNAseq library size.“ the authors did not address the association of 4th archetype for their model.
4. The PCA plots in Figures 1 and 2 should be included the sample annotation, including the sample type, expression pattern scale. It may provide better elucidation for the observations. Similarly, the PCA plots in Figure 3 needs to be revised, particularly the distribution and frequency of the alternatively splicing event.
5. In the section of Figure 4, these hypotheses for the RBP-regulation on alternative splicing in tumors require the advanced analyses. Such like, the expression pattern association of these RBP candidates with tumor burden and clinical relevance (survival and/or recurrence), and differential expression pattern between normal and tumors, are suggested included.
6. Low resolutions of motif analyses in Figure 4 needs to improved.
7. Overall, the authors made a hypothesis based on the 136 samples, whether the model could be validated in other sample size analyses. The authors should discuss this issue in discussion section.
Comments on the Quality of English LanguageIt needs to be improved.
Round 2
Reviewer 2 Report
Comments and Suggestions for Authors
The authors meticulously addressed the raised concerns.